# DEEP AUTO-DEFERRING POLICY
# FOR APPROXIMATING MAXIMUM INDEPENDENT SETS

## ABSTRACT

Designing efficient algorithms for combinatorial optimization appears ubiquitously in various scientific fields. Recently, deep reinforcement learning (DRL) frameworks have gained considerable attention as a new approach: they can automatically train a good solver while relying less on sophisticated domain knowledge of the target problem. However, the number of stages (until reaching the final solution) required by existing DRL solvers is proportional to the size of the input graph, which hurts their scalability to large-scale instances. In this paper, we seek to resolve this issue by proposing a novel design of DRL's policy, coined auto-deferring policy (AutoDP), automatically stretching or shrinking its decision process. Specifically, it decides whether to finalize the value of each vertex at the current stage or defer to determine it at later stages. We apply the proposed AutoDP framework to the maximum independent set (MIS) problem and its variants under various scenarios. Our experimental results demonstrate significant improvement of AutoDP over the current state-of-the-art DRL scheme in terms of computational efficiency and approximation quality. The reported performance of our generic DRL scheme is also comparable with that of the existing solvers for MIS, e.g., AutoDP outperforms them for the Barabási-Albert graph with two million vertices.

## 1 INTRODUCTION

Combinatorial optimization is an important mathematical field addressing fundamental questions of computation, where its popular examples include the maximum independent set (MIS, Miller & Muller 1960), satisfiability (SAT, Schaefer 1978) and traveling salesman problem (TSP, Voigt 1831). Such problems also arise in various applied fields, e.g., sociology (Harary & Ross, 1957), operations research (Feo et al., 1994) and bioinformatics (Gardiner et al., 2000). However, most combinatorial optimization problems are NP-hard to solve, i.e., exact solutions are typically intractable to find in practical situations. To alleviate this issue, there have been huge efforts in designing fast heuristic solvers (Biere et al., 2009; Knuth, 1997; Mezard et al., 2009) that generate approximate solutions for such scenarios.

Recently, the remarkable progress in deep learning has stimulated increased interest in learning such heuristics based on deep neural networks (DNNs). Such learning-based approaches are attractive since one could automate the design of approximation algorithms with less reliance on sophisticated knowledge. As the most straight-forward way, supervised learning schemes can be used for training DNNs to imitate the solutions obtained from existing solvers (Vinyals et al., 2015; Li et al., 2018; Selsam et al., 2019). However, such a direction can be criticized, for its quality and applicability are bounded by those of existing solvers. An ideal direction is to discover new solutions in a fully unsupervised manner, potentially outperforming those based on domain-specific knowledge.

To this end, deep reinforcement learning (DRL) schemes have been studied in the literature (Bello et al., 2016; Khalil et al., 2017; Deudon et al., 2018; Kool et al., 2019) as a Markov decision process (MDP) can be naturally designed with rewards derived from the optimization objective of the target problem. Then, the corresponding agent can be trained based on existing training schemes of DRL, e.g., Bello et al. (2016) trained the so-called pointer network for the TSP based on actor-critic training. Such DRL-based methods are especially attractive since they can even solve unexplored problems where domain knowledge is scarce and no efficient heuristic is known. However, the existing methods

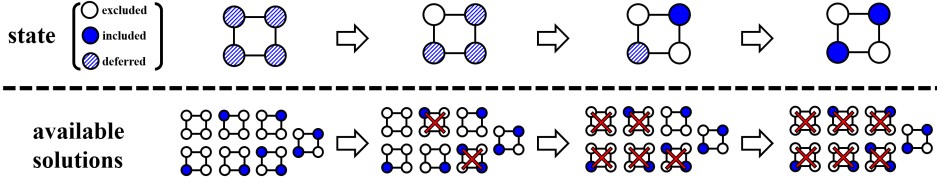

Figure 1: Illustration of the proposed Markov decision process.

struggle to compete with the existing highly optimized solvers. In particular, the gap grows larger when the problem requires solutions with higher dimensions or more complex structures.

Our motivation stems from the observation that existing DRL-based solvers lack efficient policies for generating solutions to combinatorial problems. Specifically, they are mostly based on emulating greedy iterative heuristics (Bello et al., 2016; Khalil et al., 2017) and become too slow for training on large graphs. Their choice seems inevitable since an algorithm that generates a solution based on a single feed-forward pass of DNN is potentially hard to train due to large variance in reward signals coming from high dimensional solutions.

**Contribution.** In this paper, we propose a new scalable DRL framework, coined auto-deferring policy (AutoDP), designed towards solving combinatorial problems on large graphs. We particularly focus on applying AutoDP to the MIS problem (and its variants) which attempts to find a maximum set of vertices in the graph where no pair of vertices are adjacent to each other. Our choice of the MIS problem is motivated by its hardness and applicability. First, the MIS problem is impossible to approximate in polynomial time by a constant factor (unless P=NP) (Hastad, 1996), in contrast to (Euclidean or metric) TSP which can be approximated by a factor of $1.5$ (Christofides, 1976). Next, it has wide applications including classification theory (Feo et al., 1994), computer vision (Sander et al., 2008) and communication algorithms (Jiang & Walrand, 2010).

The main novelty of AutoDP is automatically stretching the determination of the solution throughout multiple steps. In particular, the agent iteratively acts on every undetermined vertex for either (a) determining the membership of the vertex in the solution or (b) deferring the determination to be made in later steps (see Figure 1 for illustration). Inspired by the celebrated survey propagation (Braunstein et al., 2005) for solving the SAT problem, AutoDP could be interpreted as prioritizing the "easier" decisions to be made first, which in turn simplifies the harder ones by eliminating the source of uncertainties. Compared to the greedy strategy (Khalil et al., 2017) which determines the membership of a single vertex at each step, our framework brings significant speedup by allowing determinations on as many vertices as possible to happen at once.

Based on such speedup, AutoDP can solve the optimization problem by generating a large number of candidate solutions in a limited time budget, then reporting the best solution among them. In such a scenario, it is beneficial for the algorithm to generate diverse candidates. To this end, we additionally give a novel diversification bonus to our agent during training, which explicitly encourages the agent to generate a large variety of solutions. Specifically, we create a "coupling" of MDPs to generate two solutions for the given MIS problem and reward the agents for a large deviation between the solutions. The resulting reward efficiently stimulates the agent to explore high-dimensional input spaces and to improve the performance at the evaluation.

We empirically validate the AutoDP method on various types of graphs including the Erdös-Rényi (Erdős & Rényi, 1960) model, the Barabási-Albert (Albert & Barabási, 2002) model, the SATLIB (Hoos & Stützle, 2000) benchmark and real-world graphs (Hamilton et al., 2017; Yanardag & Vishwanathan, 2015; Leskovec & Sosič, 2016). Our algorithm shows consistent superiority over the existing state-of-the-art DRL method (Khalil et al., 2017) both in terms of speed and quality of the solution, and can compete with the existing MIS solver (ReduMIS, Lamm et al. 2017) under a similar time budget. For example, AutoDP even outperforms ReduMIS in the Barabási-Albert graph with two million vertices using a smaller amount of time. Furthermore, we also show that our fully learning-based scheme generalizes well even to graph types unseen during training and works well even for other variants of the MIS problem: the maximum weighted independent set (MWIS) problem and the prize collecting maximum independent set (PCMIS) problem (see Appendix B). This sheds light on its potential of being a generic solver that works for arbitrary large-scale graphs.

## 2 RELATED WORKS

The maximum independent set (MIS) problem is a prototypical NP-hard task where its optimal solution cannot be approximated by a constant factor in polynomial time (unless P = NP) (Hastad, 1996), although it admits a nearly linear factor approximation algorithm (Boppana & Halldórsson, 1992). It is also known to be a $W[1]$-hard problem in terms of fixed-parameter tractability (Downey & Fellows, 2012). Since the problem is NP-hard, existing methods (Tomita et al., 2010; San Segundo et al., 2011) for exactly solving the MIS problem often suffers from a prohibitive amount of computation in large graphs. To resolve this issue, a wide range of solvers have been developed for approximately solving the MIS problem (Grosso et al., 2008; Andrade et al., 2012; Dahlum et al., 2016; Lamm et al., 2017; Chang et al., 2017; Hespe et al., 2019). Notably, Lamm et al. (2017) developed a combination of an evolutionary algorithm with graph kernelization techniques for the MIS problem. Later, Chang et al. (2017) and Hespe et al. (2019) further improved the graph kernelization technique by introducing new reduction rules and parallelization based on graph partitioning, respectively.

In the context of solving combinatorial optimization using neural networks, Hopfield & Tank (1985) first applied the Hopfield-network for solving the traveling salesman problem (TSP). Since then, several works also tried to utilize neural networks in different forms, e.g., see Smith (1999) for a review of such papers. Such works were mostly used for solving combinatorial optimization through online learning, i.e., training was performed for each problem instance separately. More recently, (Vinyals et al., 2015) and (Bello et al., 2016) proposed to solve TSP using an attention-based neural network trained in an offline way. They showed promising results that stimulated many other works to use neural networks for solving combinatorial problems (Khalil et al., 2017; Selsam et al., 2019; Deudon et al., 2018; Amizadeh et al., 2018; Li et al., 2018; Kool et al., 2019). Importantly, Khalil et al. (2017) proposed a reinforcement learning framework for solving the minimum vertex cover problem, which is equivalent to solving the MIS problem. They query the agent for each vertex to add as a new member of the vertex cover at each step of the Markov decision process. However, such a procedure often leads to a prohibitive amount of computation on graphs with large vertex covers. Next, Li et al. (2018) aim for developing a supervised learning framework for solving the MIS problem. At an angle, their framework is similar to ours since they use hand-designed rules to defer the solution generation procedure at each step. However, it is hard to fairly compare with ours since the supervised learning scheme is highly sensitive to the quality of solutions obtained from existing solvers and is often too expensive to apply, e.g., for the MIS-variants.

## 3 DEEP AUTO-DEFERRING POLICY

In this paper, we focus on solving the maximum independent set (MIS) problem. Given a graph $\mathcal{G} = (\mathcal{V}, \mathcal{E})$ with vertices $\mathcal{V}$ and edges $\mathcal{E}$, an *independent set* is a subset of vertices $\mathcal{I} \subseteq \mathcal{V}$ where no two vertices in the subset are adjacent to each other. A solution to the MIS problem can be represented as a binary vector $\boldsymbol{x} = [x_i : i \in \mathcal{V}] \in \{0,1\}^{\mathcal{V}}$ with maximum possible cardinality $\sum_{i \in \mathcal{V}} x_i$, where each element $x_i$ indicates the membership of vertex $i$ in the independent set $\mathcal{I}$, i.e., $x_i = 1$ if and only if $i \in \mathcal{I}$. Initially, the algorithm has no assumption about its output, i.e., both $x_i = 0$ and $x_i = 1$ are possible for all $i \in \mathcal{V}$. At each iteration, the agent acts on each undetermined vertex $i$ by either (a) determining its membership to be a certain value, i.e., set $x_i = 0$ or $x_i = 1$, or (b) deferring the determination to be made later iterations. The iterations are repeated until all the membership of vertices in the independent set is determined. Such a strategy could be interpreted as progressively narrowing down the set of candidate solutions at each iteration (see Figure 1 for illustration). Intuitively, the act of deferring could be seen as prioritizing to choose the values of the "easier" vertices first. After the decisions are made, decisions on "hard" vertices become easier since the decisions on its surrounding easy vertices are better known. We additionally provide an illustration of the whole algorithm in Appendix A.

### 3.1 DEFERRED MARKOV DECISION PROCESS

To train the agent via reinforcement learning, we formulate the proposed algorithm as a Markov decision process (MDP).

**State.** Each state of the MDP is represented as a *vertex-state* vector $\boldsymbol{s} = [s_i : i \in \mathcal{V}] \in \{0, 1, *\}^{\mathcal{V}}$, where the vertex $i \in \mathcal{V}$ is determined to be excluded or included in the independent set whenever

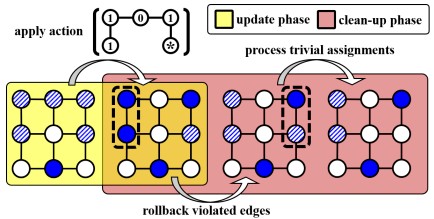

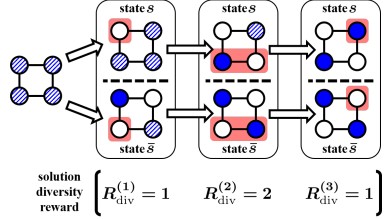

Figure 2: Illustration of the transition function with the update and the clean-up phases.

Figure 3: Illustration of coupled MDP with the corresponding solution diversity reward.

$s_i = 0$ or $s_i = 1$, respectively. Otherwise, $s_i = *$ indicates the determination has been deferred and expected to be made in later iterations. The MDP is initialized with the deferred vertex-states, i.e., $s_i = *$ for all $i \in \mathcal{V}$, and terminated when there is no deferred vertex-state left.

**Action.** Actions correspond to new assignments for the next state of vertices. Since vertex-states of included and excluded vertices are immutable, the assignments are defined only on the deferred vertices. It is represented as a vector $\boldsymbol{a}_* = [a_i : i \in \mathcal{V}_*] \in \{0, 1, *\}^{\mathcal{V}_*}$ where $\mathcal{V}_*$ denotes a set of current deferred vertices, i.e., $\mathcal{V}_* = \{i : i \in \mathcal{V}, x_i = *\}$.

**Transition.** Given two consecutive states $\boldsymbol{s}, \boldsymbol{s}'$ and the corresponding assignment $\boldsymbol{a}_*$, the transition $P_{\boldsymbol{a}_*}(\boldsymbol{s}, \boldsymbol{s}')$ consists of two deterministic phases: the *update phase* and the *clean-up phase*. In the update phase, the assignment $\boldsymbol{a}_*$ generated by the policy is updated for the corresponding vertices $\mathcal{V}_*$ to result in an intermediate vertex-state $\widehat{\boldsymbol{s}}$, i.e., $\widehat{s}_i = a_i$ if $i \in \mathcal{V}_*$ and $\widehat{s}_i = s_i$ otherwise. In the cleanup phase, the intermediate vertex-state vector $\widehat{\boldsymbol{s}}$ is modified to yield a valid vertex-state vector $\boldsymbol{s}'$, where included vertices are only adjacent to the excluded vertices. To this end, whenever there exists a pair of included vertices adjacent to each other, they are both mapped back to the deferred vertex-state. Next, any deferred vertex neighboring with an included vertex is excluded. If the state reaches the pre-defined time limit, all deferred vertices are automatically excluded. See Figure 2 for a more detailed illustration of the transition.

**Reward.** Finally, reward $R(\boldsymbol{s}, \boldsymbol{s}')$ is defined as the increase in cardinality of included vertices, i.e., $R(\boldsymbol{s}, \boldsymbol{s}') = \sum_{i \in \mathcal{V}_* \setminus \mathcal{V}'_*} s'_i$, where $\mathcal{V}_*$ and $\mathcal{V}'_*$ are the set of vertices with deferred vertex-state with respect to $\boldsymbol{s}$ and $\boldsymbol{s}'$, respectively. By doing so, the overall return of the MDP corresponds to the cardinality of the independent set returned by our algorithm.

## 3.2 TRAINING WITH DIVERSIFICATION REWARD

Next, we introduce an additional reward term for encouraging diversification of solutions generated by the agent. Such regularization is motivated by our evaluation method which samples multiple candidate solutions to report the best one as the final output. For such scenarios, it would be beneficial to generate diverse solutions of high maximum score, rather than ones of similar scores. One might argue that the existing entropy regularization (Williams & Peng, 1991) for encouraging exploration over MDP could be used for this purpose. However, the entropy regularization attempts to generate diverse trajectories of the same MDP which does not necessarily lead to diverse solutions at last, since there exist many trajectories resulting in the same solution (see Section 3.1). We instead directly maximize the diversity among solutions by a new reward term. To this end, we "couple" two copies of MDPs defined in Section 3.1 into a new MDP by sharing the same graph $\mathcal{G}$ with a pair of distinct vertex-state vectors $(\boldsymbol{s}, \bar{\boldsymbol{s}})$. Although the coupled MDPs are defined on the same graph, the corresponding agents work independently to result in a pair of solutions $(\boldsymbol{x}, \bar{\boldsymbol{x}})$. Then, we directly reward the deviation between the coupled solutions in terms of $\ell_1$-norm, i.e., $\|\boldsymbol{x} - \bar{\boldsymbol{x}}\|_1$. Similar to the original objective of MIS, it is decomposed into rewards in each iteration of the MDP defined as follows:

$$R_{\text{div}}(\boldsymbol{s}, \boldsymbol{s}', \bar{\boldsymbol{s}}, \bar{\boldsymbol{s}}') = \sum_{i \in \widehat{\mathcal{V}}} |s'_i - \bar{s}'_i|, \quad \text{where} \quad \widehat{\mathcal{V}} = (\mathcal{V}_* \setminus \mathcal{V}'_*) \cup (\bar{\mathcal{V}}_* \setminus \bar{\mathcal{V}}'_*),$$

where $(\boldsymbol{s}', \bar{\boldsymbol{s}}')$ denotes the next pair of vertex-states in the coupled MDP. One can observe that $\widehat{\mathcal{V}}$ denotes the most recently updated vertices in each MDP. In practice, such reward $R_{\text{div}}$ can be used along with the maximum entropy regularization for training the agent to achieve the best performance. See Figure 3 for an example of coupled MDP with the proposed reward.

Our algorithm is based on actor-critic training with policy network $\pi_\theta(\boldsymbol{a}|\boldsymbol{s})$ and value network $V_\phi(\boldsymbol{s})$ parameterized by the graph convolutional network (GCN, Kipf & Welling 2017). Each GCN consists of multiple layers $h_n$ with $n = 1, \cdots, N$ where the $n$-th layer with weights $\mathbf{W}_1^{(n)}$ and $\mathbf{W}_2^{(n)}$ performs the following transformation on input $\mathbf{H}$:

$$h^{(n)}(\mathbf{H}) = \mathrm{ReLU}\big(\mathbf{H}\mathbf{W}_1^{(n)} + \mathbf{D}^{-\frac{1}{2}}\mathbf{A}\mathbf{D}^{-\frac{1}{2}}\mathbf{H}\mathbf{W}_2^{(n)}\big).$$

Here $\mathbf{A}, \mathbf{D}$ correspond to adjacency and degree matrix of the graph $\mathcal{G}$, respectively. At the final layer, the policy and value networks apply softmax function and graph readout function with sum pooling (Xu et al., 2019) instead of ReLU to generate actions and value estimates, respectively. We only consider the subgraph that is induced on the deferred vertices $\mathcal{V}_*$ as the input of the networks since the determined part of the graph no longer affects the future rewards of the MDP. Features corresponding to the vertices are given as their node degrees and the current iteration-index of MDP.

To train the agent, proximal policy optimization (Schulman et al., 2017) is used. Specifically, networks are trained for maximizing the following objective:

$$\mathcal{L} := \mathbb{E}_t\bigg[ \min\bigg( \widehat{A}(\boldsymbol{s}^{(t)}) \prod_{i\in\mathcal{V}} r_i^{(t)}(\theta), \widehat{A}(\boldsymbol{s}^{(t)}) \prod_{i\in\mathcal{V}} \texttt{clip}(r_i^{(t)}(\theta), 1-\varepsilon, 1+\varepsilon) \bigg) \bigg],$$

$$r_i^{(t)}(\theta) = \frac{\pi_\theta(a_i^{(t)}|\boldsymbol{s}^{(t)})}{\pi_{\theta_{\mathrm{old}}}(a_i^{(t)}|\boldsymbol{s}^{(t)})}, \qquad \widehat{A}(s) = \sum_{t'=t}^{T}\big(R^{(t')} + R_{\mathrm{div}}^{(t')}\big) - V_\phi(s),$$

where $\boldsymbol{s}^{(t)}$ denotes the $t$-th vertex-state vector and other elements of the MDP are defined similarly. In addition, $\texttt{clip}(\cdot)$ is the clipping function for updating the agent more conservatively and $\theta_{\mathrm{old}}$ is the parameter of the policy network from the previous iteration of updates.

## 4 EXPERIMENTS

In this section, we report experimental results on the proposed auto-deferring policy (AutoDP) described in Section 3 for solving the maximum independent set (MIS) problem. We also provide evaluation of our AutoDP framework on variants of the MIS problem in Appendix B, which demonstrates that our framework is applicable to problems different from the original MIS problem. Experiments were conducted on a large range of graphs varying from small synthetic graphs to large-scale real-world graphs. We evaluated our AutoDP scheme by sampling multiple solutions and then reporting the performance of the best solution. The resulting schemes are coined AutoDP-10, AutoDP-100, and AutoDP-1000 corresponding to the number of samples chosen from $10, 100$ and $1000$, respectively. We compared our framework with the deep reinforcement learning (DRL) algorithm by Khalil et al. (2017), coined S2V-DQN, for solving the MIS problem. Note that other DRL schemes in the literature, e.g., pointer network (Bello et al., 2016), and attention layer (Kool et al., 2019) are not comparable since they are specialized to TSP-like problems. We additionally consider three conventional MIS solvers as baselines. First, we consider the theoretically guaranteed algorithm of Boppana & Halldórsson (1992) based on iterative exclusion of subgraphs, coined ES, having an approximation ratio $O(|\mathcal{V}|/(\log|\mathcal{V}|)^2)$ for the MIS problem. Next, we consider the integer programming solver IBM ILOG CPLEX Optimization Studio V12.9.0 (ILO, 2014), coined CPLEX.[1] We also consider the MIS heuristic proposed by Lamm et al. (2017), coined ReduMIS. Note that we use the implementation of ReduMIS equipped with graph kernelization proposed by Hespe et al. (2019). We additionally provide evaluation of the AudoDP framework compared to the supervised learning framework of Li et al. (2018) in Appendix C. Further details of the implementation and datasets are provided in Appendix E and F, respectively.

### 4.1 PERFORMANCE EVALUATION

We now demonstrate the performance of our algorithm along with other baselines on various datasets. First, we consider experiments on randomly generated synthetic graphs from the Erdös-Rényi (ER, Erdős & Rényi 1960) and Barabási-Albert (BA, Albert & Barabási 2002) models. Following Khalil

---

[1] Note that CPLEX is able to provide proof of optimality in addition to the solution for the MIS problem.

[2] The authors of S2V-DQN only reported experiments with respect to graphs of size up to five hundred.

Table 1: Performance evaluation on ER and BA datasets. The bold numbers indicate the best performance within the same category of algorithms. The relative differences shown in brackets are measured with respect to S2V-DQN.

| Type | $|\mathcal{V}|$ | Value | Traditional | | | DRL-based | | | | | | |
|------|------|-------|-----|-------|--------|---------|-----------|-----------|---------|-----------|---------|----------|
| | | | ES | CPLEX | ReduMIS | S2V-DQN | AutoDP-10 | | AutoDP-100 | | AutoDP-1000 | |
| ER | (15, 20) | Obj. | 7.800 | **8.844** | **8.844** | 8.840 | **8.844** | (+0.1%) | **8.844** | (+0.1%) | **8.844** | (+0.1%) |
| | | Time | 0.005 | 0.003 | 0.024 | 0.004 | 0.002 | (−60.7%) | 0.005 | (+15.6%) | 0.056 | (+1185.5%) |
| | (40, 50) | Obj. | 13.83 | **16.57** | **16.57** | 16.42 | 16.55 | (+0.7%) | **16.57** | (+0.9%) | **16.57** | (+0.9%) |
| | | Time | 0.033 | 0.062 | 12.374 | 0.015 | 0.004 | (−73.1%) | 0.016 | (+9.6%) | 0.164 | (+1002.0%) |
| | (50, 100) | Obj. | 17.01 | **21.11** | **21.11** | 20.61 | 21.04 | (+2.1%) | 21.10 | (+2.4%) | **21.11** | (+2.4%) |
| | | Time | 0.117 | 0.137 | 24.387 | 0.030 | 0.007 | (−77.8%) | 0.028 | (−4.3%) | 0.283 | (+852.6%) |
| | (100, 200) | Obj. | 21.59 | 27.87 | **27.95** | 26.27 | 27.67 | (+5.3%) | 27.87 | (+6.1%) | 27.93 | (+6.4%) |
| | | Time | 0.608 | 10.748 | 30.109 | 0.078 | 0.029 | (−63.5%) | 0.085 | (+8.5%) | 0.666 | (+748.7%) |
| | (400, 500) | Obj. | 29.28 | 35.76 | **39.83** | 35.05 | 38.29 | (+9.2%) | 39.11 | (+11.6%) | 39.54 | (+12.8%) |
| | | Time | 10.823 | 30.085 | 30.432 | 0.633 | 0.158 | (−75.1%) | 0.407 | (−35.7%) | 2.768 | (+336.8%) |
| BA | (15, 20) | Obj. | 6.06 | **7.019** | **7.019** | 7.011 | **7.019** | (+0.1%) | **7.019** | (+0.1%) | **7.019** | (+0.1%) |
| | | Time | 0.003 | 0.004 | 0.041 | 0.005 | 0.002 | (−60.2%) | 0.006 | (+19.4) | 0.064 | (+1103.4) |
| | (40, 50) | Obj. | 14.81 | **18.91** | **18.91** | 18.87 | **18.91** | (+0.2%) | **18.91** | (+0.2%) | **18.91** | (+0.2%) |
| | | Time | 0.031 | 0.020 | 0.396 | 0.013 | 0.003 | (−79.2%) | 0.011 | (−18.4%) | 0.118 | (+778.1%) |
| | (50, 100) | Obj. | 24.77 | **32.07** | **32.07** | 31.96 | **32.07** | (+0.3%) | **32.07** | (+0.3%) | **32.07** | (+0.3%) |
| | | Time | 0.130 | 0.038 | 0.739 | 0.022 | 0.003 | (−84.7%) | 0.018 | (−19.0%) | 0.199 | (+794.3%) |
| | (100, 200) | Obj. | 49.87 | **66.07** | **66.07** | 65.52 | 66.05 | (+0.8%) | **66.07** | (+0.8%) | **66.07** | (+0.8%) |
| | | Time | 0.938 | 0.088 | 2.417 | 0.047 | 0.007 | (−85.2%) | 0.038 | (−19.6%) | 0.380 | (+703.6%) |
| | (400, 500) | Obj. | 148.51 | **204.14** | **204.14** | 202.91 | 204.04 (+0.6%) | | 204.10 (+0.6%) | | **204.12** (+0.6)% | |
| | | Time | 23.277 | 0.322 | 15.080 | 0.177 | 0.024 | (−86.4%) | 0.131 | (−25.8%) | 1.111 | (+527.5%) |

Table 2: Performance evaluation on SATLIB, PPI, REDDIT, and as-Caida datasets. The bold numbers indicate the best performance within the same category of algorithms. The relative differences shown in brackets are measured with respect to S2V-DQN, except for the case of as-Caida dataset where S2V-DQN underperforms significantly.[2]

| Type | $|\mathcal{V}|$ | Value | Traditional | | DRL-based | | | | |
|------|------|-------|-------|---------|---------|----------------|----------------|----------------|
| | | | CPLEX | ReduMIS | S2V-DQN | AutoDP-10 | AutoDP-100 | AutoDP-1000 |
| SATLIB | (1209, 1347) | Obj. | 426.8 | **426.9** | 413.8 | 423.8 (+2.4%) | 424.8 (+2.7%) | **425.4** (+2.8%) |
| | | Time | 9.490 | 30.110 | 2.260 | 0.311 (−86.2%) | 1.830 (−19.0%) | 16.409 (+626.1%) |
| PPI | (591, 3480) | Obj. | **1147.5** | **1147.5** | 893.0 | 1144.5 (+28.2%) | 1146.5 (+28.4%) | **1147.0** (+28.4%) |
| | | Time | 24.685 | 30.23 | 6.285 | 0.786 (−87.5%) | 1.770 (−71.8%) | 11.469 (+82.3%) |
| REDDIT (MULTI-5K) | (22, 3648) | Obj. | **370.6** | **370.6** | 370.1 | **370.6** (+0.1%) | **370.6** (+0.1%) | **370.6** (+0.1%) |
| | | Time | 0.008 | 0.159 | 0.076 | 0.071 (−6.6%) | 0.551 (+625.0%) | 5.500 (+7136.8%) |
| REDDIT (MULTI-12K) | (2, 3782) | Obj. | **303.5** | **303.5** | 302.8 | 257.4 (−15.0%) | 292.6 (−3.4%) | **303.5** (+0.2%) |
| | | Time | 0.007 | 0.188 | 1.883 | 0.003 (−99.8%) | 0.025 (−98.7%) | 0.451 (−76.1%) |
| REDDIT (BINARY) | (6, 3782) | Obj. | **329.3** | **329.3** | 328.6 | **329.3** (+0.2%) | **329.3** (+0.2%) | **329.3** (+0.2%) |
| | | Time | 0.007 | 0.306 | 0.055 | 0.020 (−63.6%) | 0.173 (+214.6%) | 2.627 (+4676.4%) |
| as-Caida | (8020, 26 475) | Obj. | **20 049.2** | **20 049.2** | 324.0 | **20 049.2** | **20 049.2** | **20 049.2** |
| | | Time | 0.477 | 1.719 | 601.351 | 0.812 | 6.106 | 62.286 |

et al. (2017), the edge ratio of ER graphs and average degree of BA graphs are set to 0.15 and 8, respectively. Datasets are specified by their type of model and an interval for choosing the number of vertices uniformly at random, e.g., ER-(50, 100) denotes the set of ER graphs generated with the number of vertices larger than 50 and smaller than 100. Next, we consider experiments on more challenging datasets with larger sizes, namely the SATLIB, PPI, REDDIT, and as-Caida datasets constructed from SATLIB benchmark (Hoos & Stützle, 2000), protein-protein interactions (Hamilton et al., 2017), social networks (Yanardag & Vishwanathan, 2015) and road networks (Leskovec & Sosič, 2016), respectively. See the appendix for more details on the datasets. The time limit of the CPLEX and ReduMIS are set to 30 seconds on ER and BA datasets and 1800 seconds on the rest of the datasets to provide comparable baselines.[3] The corresponding results are reported in Table 1 and 2. Note that the ES method was excluded from comparison in large graphs since it required a prohibitive amount of computation.

In Table 1 and 2, one can observe that our AutoDP algorithms significantly outperform the deep reinforcement learning baseline, i.e., S2V-DQN, across all types of graphs and datasets. For example, AutoDP-10 is always able to find a better solution than S2V-DQN much faster. The gap grows larger in more challenging datasets, e.g., see Table 2. It is also impressive to observe that our algorithm can find the best solution in seven out of ten datasets in Table 1 and four out of five datasets in Table

---

[3] The solvers occasionally violate the time limit due to their pre-solving process.

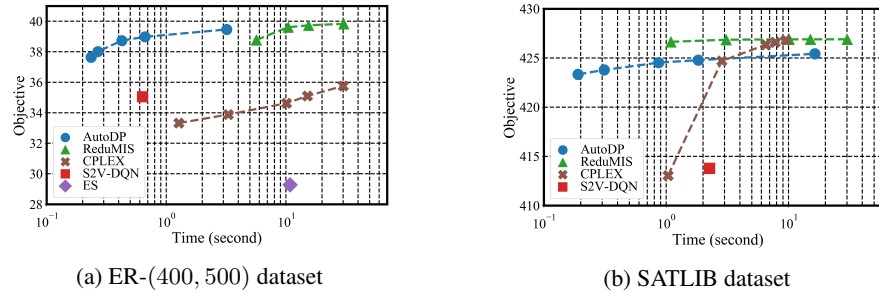

(a) ER-$(400, 500)$ dataset

(b) SATLIB dataset

Figure 4: Evaluation of trade-off between time and objective (upper-left side is of better trade-off).

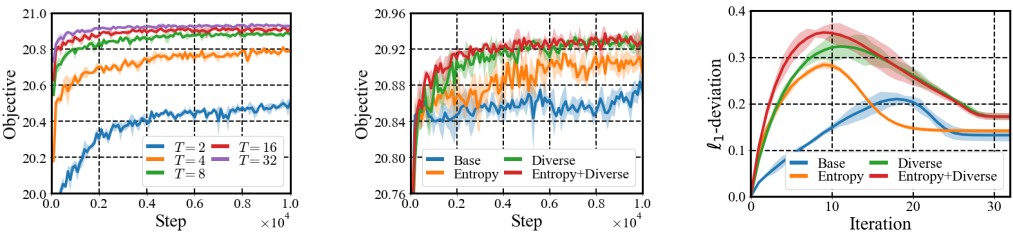

(a) Performance with varying $T$    (b) Contribution of each regularizers   (c) Deviation in intermediate stages

Figure 5: Illustration of ablation studies done on ER-$(50, 100)$ dataset. The solid line and shaded regions represent the mean and standard deviation across 3 runs respectively Note that the standard deviation in (c) was enlarged ten times for better visibility.

2. Furthermore, we observe that our algorithm achieves better objective than the CPLEX solver on ER-$(100, 200)$ and ER-$(400, 500)$ datasets, while consuming a smaller amount of time. The highly optimized ReduMIS solver tends to acquire the best solutions consistently. However, it is often worse than ours given a limited time budget, as described in what follows.

We investigate the trade-off between objective and time for algorithms in Figure 4. To this end, we evaluate algorithms on ER-$(400, 500)$ and SATLIB datasets with varying numbers of samples for AutoDP and time limits for ReduMIS and CPLEX. It is remarkable to observe that AutoDP achieves a better objective than the CPLEX solver on both datasets under reasonably limited time. Furthermore, for time limits smaller than 10 seconds, AutoDP outperforms ReduMIS on ER-$(400, 500)$ dataset.

## 4.2 ABLATION STUDY

We ablate each component of our algorithm to validate its effectiveness. We first show that "stretching" the determination with deferred MDP indeed helps for solving the MIS problem. Specifically, we experiment with varying the maximum number of iterations $T$ in MDP by $T \in \{2, 4, 8, 16, 32\}$ on ER-$(50, 100)$ dataset. Figure 5a reports the corresponding training curves. We observe that the performance of AutoDP improves whenever an agent is given more time to generate the final solution, which verifies that the deferring of decisions plays a crucial role in solving the MIS problem.

Next, we inspect the contribution of the solution diversity reward used in our algorithm. To this end, we trained agents with four options: (a) without any exploration bonus, coined Base, (b) with the conventional entropy bonus (Williams & Peng, 1991), coined Entropy, (c) with the proposed diversification bonus, coined Diverse, and (d) with both of the bonuses, coined Entropy+Diverse. The corresponding training curves for validation scores are reported in Figure 5b. We observe that the agent trained with the proposed diversification bonus outperforms other agents in terms of validation score, confirming the effectiveness of our proposed reward. One can also observe that both methods can be combined to yield better performance, i.e., Entropy+Diverse.

Finally, we further verify our claim that the maximum entropy regularization fails to capture the diversity of solutions effectively, while the proposed solution diversity reward term does. To this end, we compare the fore-mentioned agents with respect to the $\ell_1$-deviations between the coupled intermediate vertex-states $s$ and $\bar{s}$, defined as $|\{i : i \in \mathcal{V}, s_i \neq \bar{s}_i\}|$. The corresponding results are shown in Figure 5c. One can observe that the entropy regularization leads to large deviations during the intermediate stages, but converges to solutions with smaller deviations. On the contrary, agents trained on diversification rewards succeed in enlarging the deviation between the final solutions.

Table 3: Performance evaluation for large-scale graphs. Out of budget (OB) is marked for runs violating the time and the memory budget of 10 000 seconds and 32 GB RAM, respectively. The bold numbers indicate the best performance within the same category of algorithms. The relative differences shown in brackets are measured with respect to S2V-DQN.

| Type | $|\mathcal{V}|$ | Value | Traditional | | DRL-based | | | |
| | | | CPLEX | ReduMIS | S2V-DQN | AutoDP-10 | AutoDP-100 | AutoDP-1000 |
|---|---|---|---|---|---|---|---|---|
| BA | 1 000 000 | Obj.
Time | 434 896
5967.82 | **457 349**
1802.35 | OB | 457 753
324.02 | **457 772**
5112.25 | OB |
| | 2 000 000 | Obj.
Time | OB | **909 988**
4276.43 | OB | 915 553
772.18 | **915 573**
7662.87 | OB |
| Citation (Cora) | 2708 | Obj.
Time | **1451**
0.08 | **1451**
0.04 | 1393
2.57 | **1451** (+4.2%)
1.63 (−36.5%) | **1451** (+4.2%)
2.71 (+5.3%) | **1451** (+4.2%)
13.82 (+437.7%) |
| Citation (Citeseer) | 3327 | Obj.
Time | **1867**
0.08 | **1867**
0.03 | 1840
3.07 | **1867** (+1.5%)
1.80 (−41.4%) | **1867** (+1.5%)
2.74 (−10.9%) | **1867** (+1.5%)
19.00 (+518.4%) |
| Amazon (Photo) | 7487 | Obj.
Time | **2733**
38.80 | **2733**
39.02 | 725
66.53 | 2705 (+273.1%)
2.00 (−97.0%) | 2708 (+273.5%)
4.91 (−92.6%) | 2712 (+274.1%)
32.96 (−50.5%) |
| Amazon (Computers) | 13 381 | Obj.
Time | **4829**
188.61 | **4829**
61.78 | 1281
235.80 | 4773 (+272.6%)
3.02 (−98.7%) | 4782 (+273.3%)
8.63 (−96.3%) | 4783 (+273.4%)
79.19 (−66.4%) |
| Coauthor (CS) | 18 333 | Obj.
Time | **7506**
1.50 | **7506**
0.09 | 6635
197.39 | 7479 (+12.7%)
3.03 (−98.5%) | 7479 (+12.7%)
13.23 (−93.3%) | 7483 (+12.8%)
122.44 (−38.0%) |
| Coauthor (Physics) | 34 493 | Obj.
Time | 11 351
1802.80 | **11 353**
81.34 | 2156
1564.13 | 11 176 (+418.4%)
8.63 (−99.5%) | 11 186 (+418.8%)
38.71 (−97.5%) | **11 190** (+419.0%)
385.93 (−75.3%) |
| SNAP (web-Stanford) | 281 903 | Obj.
Time | 163 385
996.25 | **163 391**
44.85 | OB | 160 784
13.35 | 160 837
137.98 | **160 872**
1357.11 |
| SNAP (web-NotreDame) | 325 729 | Obj.
Time | 251 846
64.97 | **251 850**
1802.34 | OB | 250 365
11.81 | 250 384
119.48 | **250 409**
1046.48 |
| SNAP (web-BerkStan) | 685 230 | Obj.
Time | 125 194
1875.96 | **408 483**
100.17 | OB | 403 166
88.67 | 403 189
975.35 | **403 231**
9940.93 |
| SNAP (soc-Pokec) | 1 632 803 | Obj.
Time | OB | **788 907**
1805.95 | OB | 784 843
770.55 | **784 891**
7512.06 | OB |
| SNAP (wiki-topcats) | 1 791 489 | Obj.
Time | OB | **986 180**
1060.09 | OB | 958 980
850.80 | **959 051**
9121.85 | OB |

## 4.3 GENERALIZATION TO UNSEEN GRAPHS

Now we examine the potential of our method as a generic solver, i.e., whether the algorithm's performance generalizes well to graphs unseen during training. To this end, we train AutoDP and S2V-DQN models on BA-$(400, 500)$ dataset and evaluate them on the following real-world graph datasets: Coauthor, Amazon (Shchur et al., 2018) and Stanford Network Analysis Platform (SNAP, Leskovec & Sosič 2016). We additionally evaluate on BA graphs with millions of vertices. We also evaluate the generalization of the algorithm across synthetic graphs with different types and sizes in Appendix D. Similar to the experiments in Table 2, we set the time limit on CPLEX and ReduMIS by 1800 seconds. The ES method is again excluded as being computationally prohibitive to compare. As reported in Table 3, AutoDP successfully generates solutions for large scale instances which scale up to two million (2M), even though it was trained on graphs of size smaller than five hundred vertices. Most notably, AutoDP-10 outperforms the ReduMIS (state-of-the-art solver) in BA graph with 2M vertices, but six times faster. It also outperforms the CPLEX in the graphs with more than 0.5M vertices, indicating better scalability of our algorithm. Note that CPLEX also fails to generate solutions on graphs with more than 1M vertices. Such a result strongly supports the potential of AutoDP being a generic solver that could be used in place of conventional solvers. On the contrary, we found that S2V-DQN does not generalize well to large graphs: it performs worse and takes much more time to generate solutions as it requires the number of decisions proportional to the graph size.

## 5 CONCLUSION

In this paper, we propose a new reinforcement learning framework for the maximum independent set problem that is scalable to large graphs. Our main contribution is the auto-deferring policy, which allows the agent to defer the decisions on vertices for efficient expression of complex structures in the solutions. Through extensive experiments, our algorithm shows performance that is both superior to the existing reinforcement learning baseline and competitive with the conventional solvers.

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

## A   GRAPHICAL ILLUSTRATION OF AUTODP

Figure 6: Illustration of the overall AutoDP framework.

## B   VARIANTS OF THE MAXIMUM INDEPENDENT SET PROBLEM

In this section, we provide experimental results of the proposed AutoDP framework on variants of the original MIS problem. From the empirical results, we show that the AutoDP framework is flexible enough to be trained even when some settings of the original MIS problem are modified. To this end, we consider two variants of the MIS problem: the maximum weighted independent set (MWIS) problem and the prize collecting maximum independent set (PCMIS) problem. We compare our algorithm to the generic integer programming solver CPLEX. Note that the ReduMIS and the ES considered in Section 4 are unavailable for comparison on such variants of the MIS problem. Experiments are conducted on the ER graphs with varying sizes of graphs as in Section 4.1.

Table 4: Performance evaluation on the MWIS problem for the ER datasets. The bold numbers indicate the algorithm with the best performance. The relative differences shown in brackets are measured with respect to CPLEX.

| Type | $|\mathcal{V}|$ | Value | CPLEX | AutoDP-10 | AutoDP-100 | AutoDP-1000 |
|---|---|---|---|---|---|---|
| ER | (15, 20) | Obj. | **9.00** | **9.00** $(-0.0\%)$ | **9.00** $(-0.0\%)$ | **9.00** $(-0.0\%)$ |
| | | Time | 0.013 | 0.003 $(-76.9\%)$ | 0.006 $(-53.8\%)$ | 0.045 $(+246.2\%)$ |
| | (40, 50) | Obj. | **16.86** | 16.82 $(-0.2\%)$ | **16.86** $(-0.0\%)$ | **16.86** $(-0.0\%)$ |
| | | Time | 0.083 | 0.006 $(-92.8\%)$ | 0.019 $(-77.1\%)$ | 0.168 $(+102.4\%)$ |
| | (50, 100) | Obj. | **21.46** | 21.29 $(-0.8\%)$ | 21.44 $(-0.1\%)$ | **21.46** $(-0.0\%)$ |
| | | Time | 0.159 | 0.013 $(-91.8\%)$ | 0.038 $(-76.1\%)$ | 0.316 $(+98.7\%)$ |
| | (100, 200) | Obj. | 28.41 | 28.09 $(-1.1\%)$ | 28.39 $(-0.1\%)$ | **28.45** $(+0.1\%)$ |
| | | Time | 12.357 | 0.040 $(-99.7\%)$ | 0.098 $(-99.2\%)$ | 0.699 $(-94.3\%)$ |
| | (400, 500) | Obj. | 37.66 | 38.08 $(+1.1\%)$ | 39.21 $(+4.1\%)$ | **39.99** $(+6.2\%)$ |
| | | Time | 30.083 | 0.277 $(-99.1\%)$ | 0.537 $(-98.2\%)$ | 3.025 $(-89.9\%)$ |

Table 5: Performance evaluation on the PCMIS problem for the ER datasets. The bold numbers indicate the algorithm with the best performance. The relative differences shown in brackets are measured with respect to CPLEX.

| Type | $|\mathcal{V}|$ | Value | CPLEX | AutoDP-10 | AutoDP-100 | AutoDP-1000 |
|---|---|---|---|---|---|---|
| ER | (15, 20) | Obj. | **10.54** | 10.53 $(-0.1\%)$ | 10.53 $(-0.1\%)$ | 10.53 $(-0.1\%)$ |
| | | Time | 0.014 | 0.002 $(-85.7\%)$ | 0.004 $(-71.4\%)$ | 0.019 $(+35.7\%)$ |
| | (40, 50) | Obj. | **17.94** | 17.64 $(-1.7\%)$ | 17.68 $(-1.4\%)$ | 17.71 $(-1.3\%)$ |
| | | Time | 0.217 | 0.004 $(-98.2\%)$ | 0.006 $(-97.2\%)$ | 0.032 $(-85.3\%)$ |
| | (50, 100) | Obj. | **22.64** | 21.66 $(-4.0\%)$ | 21.77 $(-3.8\%)$ | 21.84 $(-3.5\%)$ |
| | | Time | 1.829 | 0.002 $(-99.9\%)$ | 0.008 $(-99.5\%)$ | 0.069 $(-96.2\%)$ |
| | (100, 200) | Obj. | 27.12 | 29.17 $(+7.6\%)$ | 29.56 $(+9.0\%)$ | **29.69** $(+9.5\%)$ |
| | | Time | 29.837 | 0.019 $(-99.9\%)$ | 0.107 $(-99.6\%)$ | 0.958 $(-96.8\%)$ |
| | (400, 500) | Obj. | 4.56 | 37.17 $(+715.1\%)$ | 38.39 $(+741.8\%)$ | **39.41** $(+764.3\%)$ |
| | | Time | 30.122 | 0.068 $(-99.7\%)$ | 0.421 $(-98.6\%)$ | 3.956 $(-86.9\%)$ |

## B.1 MAXIMUM WEIGHTED INDPENDENT SET PROBLEM

First, we report experimental results on solving the MWIS problem. Consider a graph $\mathcal{G} = (\mathcal{V}, \mathcal{E})$ associated with positive weight function $w : \mathcal{V} \to \mathbb{R}^+$. The goal of the MWIS problem is to find the independent set $\mathcal{I} \subseteq \mathcal{V}$ where the total sum of weight $\sum_{i \in \mathcal{I}} w(i)$ is maximum. Similar to the original MIS problem, the MWIS problem has many applications including signal transmission, information retrieval and computer vision (Balas & Yu, 1986). In order to apply the AutoDP framework to the MWIS problem, we simply include the weight of each vertex as its feature to the policy network and modify the reward function by the increase in weight of included vertices, i.e., $R(\boldsymbol{s}, \boldsymbol{s}') = \sum_{i \in \mathcal{V}_* \setminus \mathcal{V}'_*} s'_i w(i)$. The weights are randomly sampled from a normal distribution with mean and standard deviation fixed to $1.0$ and $0.1$, respectively. We report the corresponding results in Table 4. Here, one observes that AutoDP always achieves performance at least as good as that of CPLEX. It even achieves a better objective than CPLEX on the ER-$(100, 200)$ and ER-$(400, 500)$ datasets while using a smaller amount of time. Such a result confirms that our framework is generalizable to the WMIS problem.

Table 6: Additional evaluation of the supervised learning (SL) based framework on ER datasets. The bold numbers indicate the best performance within the same category of algorithms. The relative differences shown in brackets are measured with respect to the best performing SL-based model.

| Type | $|\mathcal{V}|$ | Value | SL-based | | DRL-based | | | |
|---|---|---|---|---|---|---|---|---|
| | | | GTS-ER | GTS-SATLIB | S2V-DQN | AutoDP-10 | AutoDP-100 | AutoDP-1000 |
| ER | (15, 20) | Obj. | 8.829 | **8.838** | 8.840 | **8.844** (+0.1%) | **8.844** (+0.1%) | **8.844** (+0.1%) |
| | | Time | 1.707 | 1.749 | 0.004 | 0.002 (−99.9%) | 0.005 (−99.7%) | 0.056 (−96.8%) |
| | (40, 50) | Obj. | 16.46 | **16.47** | 16.42 | 16.55 (+0.5%) | **16.57** (+0.6%) | **16.57** (+0.6%) |
| | | Time | 2.660 | 3.583 | 0.015 | 0.004 (−99.9%) | 0.016 (−99.6%) | 0.164 (−95.4%) |
| | (50, 100) | Obj. | 20.70 | **20.83** | 20.61 | 21.04 (+1.0%) | 21.10 (+1.2%) | **21.11** (+1.3%) |
| | | Time | 3.222 | 3.914 | 3.767 | 0.007 (−99.8%) | 0.028 (−99.4%) | 0.283 (−92.8%) |
| | (100, 200) | Obj. | **26.17** | 26.11 | 26.27 | 27.67 (+5.6%) | 27.87 (+6.5%) | **27.93** (+6.7%) |
| | | Time | 6.846 | 4.964 | 0.078 | 0.029 (−99.6%) | 0.085 (−98.8%) | 0.666 (−90.3%) |
| | (400, 500) | Obj. | **36.48** | 34.99 | 35.05 | 38.29 (+5.0%) | 39.11 (+7.2%) | **39.54** (+8.4%) |
| | | Time | 10.598 | 8.609 | 0.633 | 0.158 (−98.5%) | 0.407 (−96.2%) | 2.768 (−73.9%) |

### B.2 PRIZE COLLECTING MAXIMUM INDEPENDENT SET PROBLEM

Next, we introduce the PCMIS problem. To this end, consider a graph $\mathcal{G} = (\mathcal{V}, \mathcal{E})$ and a subset of vertices $\widetilde{\mathcal{I}} \subseteq \mathcal{V}$. Then PCMIS problem is associated with the following the "prize" function $f$ to maximize:

$$f(\widetilde{\mathcal{I}}) := |\widetilde{\mathcal{I}}| - \lambda |\{\{i, j\} : i, j \in \mathcal{I}, i \neq j\}|,$$

where $\lambda > 0$ is the penalty function for including two adjacent vertices. We set $\lambda = 0.5$ in the experiments. Such a problem could be interpreted as relaxing the hard constraints on independent set to a penalty function in the MIS problem. Especially, one can examine that optimal solution of the PCMIS problem becomes the maximum independent set when $\lambda > 1$. The PCMIS problem also corresponds to an instance of the generalized minimum vertex cover problem (Hassin & Levin, 2006). For applying the AutoDP framework on the PCMIS problem, we remove the clean-up phase in the transition function of MDP and modify the reward function $R(\boldsymbol{s}, \boldsymbol{s}')$ as the increase in prize function at each iteration, expressed as follows:

$$R(\boldsymbol{s}, \boldsymbol{s}') := \sum_{i \in \mathcal{V}_* \setminus \mathcal{V}_*'} s_i' - \lambda \sum_{i \in \mathcal{V}_* \setminus \mathcal{V}_*'} \left( \sum_{j \in \mathcal{V}_* \setminus \mathcal{V}_*' \setminus \{i\}} \frac{1}{2} s_i' s_j' + \sum_{j \in \mathcal{V} \setminus \mathcal{V}_*} s_i' s_j' \right).$$

We report the corresponding results in Table 5. Here, one observes that AutoDP underperforms compared to the CPLEX at smaller graphs, but eventually outperforms it on ER-(100, 200) and ER-(400, 500) datasets. Especially, CPLEX shows underwhelming performance on the ER-(400, 500) dataset. We hypothesize this observation to arise from the fact that the PCMIS problem is modeled as a hard integer quadratic programming. Note that the MIS problem was previously modeled by CPLEX as an integer linear programming, which is easier to solve.

## C  ADDITIONAL COMPARISON TO SUPERVISED LEARNING

In this section, we additionally report the performance of the supervised learning framework for the MIS problem proposed by Li et al. (2018), coined GTS, evaluated on the ER datasets used in Section 4.1. To this end, two types of models trained from the GTS framework are considered. First, we consider the GTS trained on ER graphs, coined GTS-ER. For supervision, we generate solutions of 2000 graphs from the corresponding ER datasets using ReduMIS. Note that the training scheme for GTS-ER was designed to match the computational requirement of GTS-ER and AutoDP. For example, it takes 16 and 8 hours to generate solutions and train the models for GTS-ER on the ER-(400, 500) dataset. AutoDP takes 24 hours to train the model on the same dataset. In addition, we consider the model trained on the SATLIB dataset with 38 000 graphs, coined GTS-SATLIB. This model was obtained from the code provided by Li et al. (2018).[4] We use the default hyperparameters from the

---

[4] https://github.com/intel-isl/NPHard

Table 7: Generalization performance of AutoDP-1000 across synthetic graphs with varying types and sizes. Rows and columns correspond to datasets used for training and evaluating the model, respectively.

| Type | $|\mathcal{V}|$ | ER | | | | BA | | | |
|---|---|---|---|---|---|---|---|---|---|
| | | (40, 50) | (50, 100) | (100, 200) | (400, 500) | (40, 50) | (50, 100) | (100, 200) | (400, 500) |
| ER | (15, 20) | **16.57** | 21.06 | 26.50 | 30.36 | **18.91** | **32.07** | 66.00 | 200.96 |
| | (40, 50) | **16.57** | 21.11 | 27.88 | 36.95 | **18.91** | **32.07** | **66.07** | 204.09 |
| | (50, 100) | - | **21.11** | 27.94 | 38.20 | - | **32.07** | **66.07** | 204.07 |
| | (100, 200) | - | - | 27.93 | 39.39 | - | - | **66.07** | 204.08 |
| | (400, 500) | - | - | - | **39.54** | - | - | - | 203.90 |
| BA | (15, 20) | **16.57** | 21.10 | 27.28 | 33.87 | **18.91** | **32.07** | 66.05 | 202.47 |
| | (40, 50) | **16.57** | 21.11 | 27.89 | 37.01 | **18.91** | **32.07** | **66.07** | 204.03 |
| | (50, 100) | - | **21.11** | 27.89 | 37.01 | - | **32.07** | **66.07** | 204.11 |
| | (100, 200) | - | - | 27.86 | 36.52 | - | - | **66.07** | 204.11 |
| | (400, 500) | - | - | - | 35.31 | - | - | - | **204.12** |

authors, e.g., graph convolutional networks are built with 20 layers and channel size of 32. For the evaluation of models from GTS, the performance of the best solutions among 1000 samples were reported. We also note that Li et al. (2018) optionally introduced a "classic element" of introducing local search and graph reduction between intermediate decisions. Although such an idea is applicable to all of GTS, AutoDP and S2V-DQN, we disable it to compare the vanilla performance of the supervised and the reinforcement learning frameworks.

The corresponding comparisons are reported in Table 6. Here, one observes that AutoDP performs better than the GTS algorithms even though they additionally require solutions to the MIS problem on the training graphs. It is also interesting to observe that GTS-SATLIB underperforms compared to GTS-ER in the ER-$(100, 200)$ and the ER-$(400, 500)$ datasets even though they were trained using much more graphs. We hypothesize that such a gap comes from the GTS-SATLIB failing to generalize on unseen graphs.

## D  GENERALIZATION BETWEEN SYNTHETIC GRAPHS

In this section, we report experiments on generalization between ER and BA graphs, where we evaluate the generalization capability of our method on different types and sizes of graphs from the training dataset. As shown in Table 7, our algorithm generalizes excellently across different sizes of graphs, e.g., the model trained on BA-$(50, 100)$ dataset achieves the best performance even on BA-$(400, 500)$ dataset. On the other side, models evaluated on unseen types of graphs tend to work slightly worse as expected. However, the results are still remarkable, e.g., the model trained on ER-$(40, 50)$ dataset almost achieves the best score in BA-$(400, 500)$ dataset.

## E  IMPLEMENTATION DETAILS

In this section, we provide additional details for our implementation of the experiments.

**Normalization of feature and reward.** The iteration-index of MDP used for input of the policy and value networks was normalized by the maximum number of iterations. Furthermore both the MIS objective and the solution diversification rewards were normalized by maximum number of vertices in the corresponding dataset.

**Hardware.** Computations for our method and S2V-DQN were done on an NVIDIA RTX 2080Ti GPU and an NVIDIA TITAN X Pascal GPU, respectively. Experiments for ES, CPLEX, and ReduMIS were run in AWS EC2 c5 instances with Intel Xeon Platinum 8124M CPU. We additionally let the CPLEX use 16 CPU cores as it allows multi-processing.

**Hyper-parameter.** Every hyper-parameter was optimized on a per graph type basis and used across all sizes within each graph type. Throughout every experiment, policy and value networks were parameterized by graph convolutional network with 4 layers and 128 hidden dimensions. Every instance of the model was trained for 20000 updates of proximal policy optimization (Schulman

et al., 2017), based on the Adam optimizer with a learning rate of 0.0001. The validation dataset was used for choosing the best performing model while using 10 samples for evaluating the performance. Reward was not decayed throughout the episodes of the Markov decision process. Gradient norms were clipped by a value of 0.5. We further provide details specific to each type of datasets in Table 8. For the compared baselines, we used the default hyper-parameters provided in the respective codes.

Table 8: Choice of hyperparameters for the experiments on performance evaluation. The REDDIT column indicates hyperparameters used for the REDDIT (BINARY, MULTI-5K, MULTI-12K) datasets.

| Parameters | ER | BA | SATLIB | PPI | REDDIT | as-Caida |
|---|---|---|---|---|---|---|
| Maximum iterations per episode | 32 | 32 | 128 | 128 | 64 | 128 |
| Number of unrolling iteration | 32 | 32 | 128 | 128 | 64 | 128 |
| Number of environments (graph instances) | 32 | 32 | 32 | 10 | 64 | 1 |
| Batch size for gradient step | 16 | 16 | 8 | 8 | 16 | 8 |
| Number of gradient steps per update | 4 | 4 | 8 | 8 | 16 | 8 |
| Solution diversity reward coefficient | 0.1 | 0.1 | 0.01 | 0.1 | 0.1 | 0.1 |
| Maximum entropy coefficient | 0.1 | 0.1 | 0.01 | 0.001 | 0.0 | 0.1 |

**Baselines.** We implemented the S2V-DQN algorithm based on the code (written in C++) provided by the authors.[5] For ER and BA models, S2V-DQN was unstable to be trained on graphs of size from $(100, 200)$ and $(400, 500)$ without pre-training. Instead, we performed fine-tuning as mentioned in the original paper (Khalil et al., 2017). First, for the ER-$(100, 200)$ and BA-$(100, 200)$ datasets, we fine-tuned the model trained on ER-$(50, 100)$ and BA-$(50, 100)$, respectively. Next, for the ER-$(400, 500)$ and BA-$(400, 500)$ datasets, we performed "curriculum learning", e.g., a model was first trained on the ER-$(50, 100)$ dataset, then fine-tuned on the ER-$(100, 200)$, ER-$(200, 300)$, ER-$(300, 400)$ and ER-$(400, 500)$ in sequence. Finally, for training S2V-DQN on large graphs used in Table 2, we were unable to train on the raw graph under available computational budget. Hence we trained S2V-DQN on subgraphs sampled from the training graphs. To this end, we sampled edges from the model uniformly at random without replacement, until the number of vertices reach 300. Then we used the subgraph induced from the sampled vertices. We run ES algorithm based on NetworkX package.[6] Next, we use CPLEX (ILO, 2014) provided on the official homepage.[7] We set the optimality gap used for the stopping criterion to $10^{-4}$. For comparison, we also report the performance of CPLEX for other values of the optimality gap is in Table 9. Finally, we use the ReduMIS algorithm implemented in the code provided by the authors.[8] Note that our implementation of ReduMIS is different from the original work (Lamm et al., 2017) since it employs a better graph kernelization technique (Hespe et al., 2019).

# F    DATASET DETAILS

In this section, we provide additional details on the datasets used for the experiments.

**ER and BA datasets.** For the ER and BA datasets, we train on graphs randomly generated on the fly and perform validation and evaluation on a fixed set of 1000 graphs.

**SATLIB dataset.** The SATLIB dataset is a popular benchmark for evaluating SAT algorithms. We specifically use the synthetic problem instances from the category of random 3-SAT instances with controlled backbone size (Singer et al., 2000). Next, we describe the procedure for reducing the SAT instances to MIS instances. To this end, a vertex is added to the graph for each literal of the SAT instance. Then edges are added for each pair of vertices satisfying the following conditions: (a) that are in the same clause or (b) they correspond to the same literals with different signs. Consequently,

---

[5]https://github.com/Hanjun-Dai/graph_comb_opt
[6]https://networkx.github.io/
[7]https://www.ibm.com/products/ilog-cplex-optimization-studio
[8]http://algo2.iti.kit.edu/kamis/

Table 9: Performance comparison of CPLEX with different optimality gap used for the stopping criterion.

| Type | $|\mathcal{V}|$ | Value | Optimality gap | | | |
|------|------|------|------|------|------|------|
| | | | $10^{-4}$ | $10^{-3}$ | $10^{-2}$ | $10^{-1}$ |
| ER | (15, 20) | Obj. | **8.84** | **8.84** | **8.84** | **8.84** |
| | | Time | 0.003 | 0.003 | 0.004 | 0.003 |
| | (40, 50) | Obj. | **16.57** | **16.57** | **16.57** | 16.43 |
| | | Time | 0.062 | 0.064 | 0.063 | 0.043 |
| | (50, 100) | Obj. | **21.11** | **21.11** | **21.11** | 20.938 |
| | | Time | 0.137 | 0.141 | 0.141 | 0.138 |
| | (100, 200) | Obj. | **27.87** | **27.87** | **27.87** | 27.81 |
| | | Time | 10.748 | 11.008 | 10.917 | 9.888 |
| | (400, 500) | Obj. | **35.76** | 35.72 | 35.73 | 35.74 |
| | | Time | 30.085 | 30.090 | 30.083 | 30.089 |

Table 10: Number of nodes, edges and graphs for each dataset used in the Table 2. Number of graphs is expressed as a tuple of the numbers of training, validation and test graphs, respectively.

| Dataset | Number of nodes | Number of edges | Number of graphs |
|------|------|------|------|
| SATLIB | (1209, 1347) | (4696, 6065) | (38 000, 1000, 1000) |
| PPI | (591, 3480) | (3854, 53 377) | (20, 2, 2) |
| REDDIT (BINARY) | (6, 3782) | (4, 4071) | (1600, 200) |
| REDDIT (MULTI-5K) | (22, 3648) | (21, 4783) | (4001, 499, 499) |
| REDDIT (MULTI-12K) | (2, 3782) | (1, 5171) | (9545, 1192, 1192) |
| as-Caida | (8020, 26 475) | (36 406, 106 762) | (108, 12, 12) |

the MIS in the resulting graph corresponds to the truth assignment to the optimal assignments of the SAT problem (Dasgupta et al., 2008).

**PPI dataset.** The PPI dataset is the protein-protein-interaction dataset with vertices representing proteins and edges representing interactions between them.

**REDDIT datasets.** The REDDIT (BINARY, MULTI-5K, MULTI-12K) datasets are constructed from online discussion threads in reddit[9] where vertices represent users and edges mean at least one of two users responded to the other user's comment.

**Autonomous system dataset.** The as-Caida dataset is a set of autonomous system graphs derived from a set of RouteViews BGP table snapshots (Leskovec et al., 2005).

**Citation dataset.** The Cora and the Citeseer are networks constructed by vertices and edges representing documentation and citation links between them, respectively (Sen et al., 2008).

**Amazon dataset.** The Computers and Photo graphs are segmented from the Amazon co-purchase graph (McAuley et al., 2015), where vertices correspond to goods and edges represent goods which are frequently purchased together.

**Coauthor dataset.** The CS and Physics graphs represent authors and the corresponding co-authorships by vertices and edges, respectively. It was collected from Microsoft Academic Graph from the KDD Cup 2016 challenge3.[10]

---

[9] https://www.reddit.com/
[10] https://kddcup2016.azurewebsites.net/

Table 11: Number of nodes and edges for each dataset used in the Table 3.

| Dataset | Number of nodes | Number of edges |
|---|---|---|
| Citeseer | 3327 | 3668 |
| Cora | 2708 | 5069 |
| Pubmed | 19 717 | 44 324 |
| Coauthor CS | 18 333 | 81 894 |
| Coauthor Physics | 34 493 | 247 962 |
| Amazon Computers | 13 381 | 245 778 |
| Amazon Photo | 7487 | 119 043 |
| web-Stanford | 281 903 | 2 312 497 |
| web-NotreDame | 325 729 | 1 497 134 |
| web-BerkStan | 685 230 | 7 600 595 |
| soc-Pokec | 1 632 803 | 30 622 564 |
| wiki-topcats | 1 791 489 | 28 511 807 |
| BA-1M | 1 000 000 | 3 999 984 |
| BA-2M | 2 000 000 | 7 999 984 |

**Web network datasets.** The web-NotreDame, web-Stanford, and web-BerkStan are graphs with vertices represent web-pages obtained from the University of Notre Dame, Stanford University, and Berkeley & Stanford University domains, respectively (Leskovec & Sosič, 2016). Edge is added for each pair of web-pages with links between them.

**Social network dataset.** The soc-Pokec is a graph representing friendships between users from a social network in Slovakia (Takac & Zabovsky, 2012).

**Wikipedia network dataset.** The wiki-topcats graph is constructed by collecting connected pages which belong to top categories containing at least 100 pages, starting from the most connected component of Wikipedia (Klymko et al., 2014).

We further provide the statistics of the dataset used in experiments corresponding to Table 2 and 3 in Table 10 and 11, respectively.

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
