# OpenReview forum: "Deep Auto-Deferring Policy for Combinatorial Optimization"
_ICLR.cc/2020/Conference — Reject_

### Official Review · AnonReviewer3 · 2019-10-20
**Official Blind Review #3**

**Rating:** 3

**Review:**

This paper aims at solving graph-based combinatorial optimization problems using a new paradigm for Deep Reinforcement Learning. In contrast with standard Markov Decision Processes used for combinatorial DRL, the authors advocate the use of “deferred” MDPs capturing more complex actions (which can choose a subset of nodes), and more sophisticated transitions, which combine an update phase together with a cleanup phase. For the DRL architecture, an Actor-Critic framework is trained using a proximal policy optimization approach. The paper specifically focuses on the Maximum Independent Set (MIS) problem. Comparative results on various instances reveal that this approach clearly outperforms the S2V-DQN algorithm and is competitive with some usual combinatorial solvers (CPLEX, ReduMIS).

The overall approach is interesting and the experimental results look promising, but it is quite difficult to accept the paper in its current state due to the following reasons:

(a) Though the paper is comprehensible, it is littered with spelling mistakes (just take the first sentence: problem -> problems). For the camera-ready version, it would be nice to use a spell/grammar checker.

(b) The authors focus on the MIS problem, but in the introduction, they claim that their approach can be applied to “any” combinatorial optimization task. Well, that is a bit of an overstatement, due to the immense variety of NP-hard optimization tasks! On the one hand, the MIS problem is already taking an important place in many areas of computer science, and I would suggest concentrating on this task - by changing the title and the summary of the paper, and by just pointing out in the conclusion that the present DRL approach might be applicable to other problems. On the other hand, if the authors are convinced that their DRL approach is generic, then the paper should include other well-known APX-hard problems such as, for example, Min Feedback Node Set, Min Balanced Cut, or Min Set Cover.

(c) The paper is missing a related-work section. To this point, the MIS problem should be presented in more detail with some theoretical results about its approximability (to within a linear factor, as shown by Boppana and Halldorsson, 1992), and its fixed-parameter tractability (it is W[1]-hard). From an algorithmic viewpoint, the ReduMIS technique is mentioned, but there are many other heuristic approaches and kernelization methods. Notably, OnlineMIS (Dahlum et. al. ESA’16) is orders of magnitude faster than ReduMIS on various large instances. Furthermore, the kernelization technique developed by Chang et. al. (SIGMOD’17) and its parallel version (Hespe et. al., JEA’19) is known to be much more efficient than the kernelization algorithm (VC-Solver) suggested in ReduMIS. Besides heuristic algorithms, very little is said about deep learning approaches for solving MIS. To this point, a more detailed presentation of  Dai et. al.’s approach  (NIPS-17) would be relevant, by explaining the similarities and differences with your approach. Finally, nothing is said about the recent deep architecture proposed by Li et. al. (NIPS’18) for solving MIS. Though they adopt a supervised learning approach, their GCN architecture shares strong similarities with the Actor-Critic framework in the present paper.

(d) The deep auto-deferring policy, presented in Section 3 looks interesting, but it is quite unclear. The main idea lies in deferred MDPs, for which the size of the action space is in $\Omega(3^N)$ because any subset of nodes can be chosen at each state. For such huge MDPs, it is not clear that we can converge to a stable policy in polynomial time. Furthermore, the authors are mixing Actor-Critic (using a graph convolution net) and Proximal Policy Optimization, together with diversification rewards, which makes the overall framework unintelligible. A graphical representation for explaining this complex framework would be welcome. Furthermore, I don’t see how the notion of diversification reward is implemented in the PPO approach for training the agent.

(e) In the experiments, some competitors are missing. As mentioned above, ReduMIS is nowadays dominated by more efficient kernelization/heuristic techniques for solving MIS. I would suggest to instead use OnlineMIS. Moreover, the choice of S2V-DQN is relevant as an RL competitor, but it would be interesting to also use the GCN architecture (Li et. al., NIPS’18) as a supervised competitor. Finally, the SMT solver Z3 is known to be quite efficient for MIS instances (as already observed by Li et. al.), and a comparison with it would indicate whether ADP is competitive (or not) with the best generic solvers.






**Experience Assessment:**

I have read many papers in this area.

**Review Assessment: Checking Correctness Of Derivations And Theory:**

I assessed the sensibility of the derivations and theory.

**Review Assessment: Checking Correctness Of Experiments:**

I assessed the sensibility of the experiments.

**Review Assessment: Thoroughness In Paper Reading:**

I read the paper at least twice and used my best judgement in assessing the paper.

---

> ### Author Response · Authors · 2019-11-14
> **Response to R3 (1/2)**
>
> Thank you for your valuable efforts and time spent on reading our paper. We believe our framework proposes an interesting idea (as pointed out by you and R1) with promising empirical results (as pointed out by you and R2), based on clear writing (as pointed out by R1 and R2). In the revised draft, we added a description of related works (in Section 2), a graphical illustration of our framework (in Appendix A) and additional experiments (in Appendix B, C, D) to incorporate all reviewers’ comments. Any modifications from the original draft are highlighted by “red” texts. We think the revised draft has much improved by clarifying the contributions, describing the related works and validating the empirical evaluation, thanks to your comments. Our responses to all your questions are provided below.
>
> Q1. It would be nice to use a spelling or grammar checker.
>
> A1. Thank you for your feedback. We have run the grammar checking software on the revised draft and removed any possible typos.
>
> Q2. It is an overstatement to claim the framework to be applicable to any combinatorial problems. The paper should (a) focus the paper on the maximum independent set (MIS) problem or (b) include other APX-hard combinatorial optimization problems.
>
> A2. Thank you very much for the valuable suggestion regarding the direction of our paper. Following your suggestion (a), we have revised the whole draft (including the title) towards focusing on solving the MIS problem. Under the decision, we also provide experimental results on applying our framework for two variants of the MIS problem in Appendix B: the maximum weighted independent set (MWIS) problem and the prize collecting maximum independent set (PCMIS) problem. In the new applications, our framework shows its general applicability by achieving better performance than the generic integer programming solver, i.e., CPLEX. Note that ReduMIS and ES are not applicable to the considered variants of the MIS problem, which highlights the advantage of our scheme.
>
> Q3. Related-work section is needed to be included in the paper.
>
> A3. Thank you for such a constructive and informative suggestion. Following your suggestion, we have added Section 2 of the revised draft for describing the related works. We included all the papers that were mentioned in your comments.
>
> Q4. It is not clear whether a policy with exponentially large action space converges in polynomial time.
>
> A4. Thank you for making an important point. While it is non-trivial to prove the polynomial convergence of our policy, we empirically observe it converge very fast. Specifically, our policy usually takes 10 000 steps to converge, e.g., see Figure 5 (a) in the revised draft. This requirement is similar to the convergence reported for S2V-DQN [1] that has a polynomially small action space.
>
> [1] Elias B. Khalil, Hanjun Dai, Yuyu Zhang, Bistra Dilkina and Le Song, “Learning combinatorial optimization algorithms over graphs”, In NeurIPS, 2017

---

> > ### Author Response · Authors · 2019-11-14
> > **Response to R3 (2/2)**
> >
> > Q5. A graphical representation of the whole framework would be helpful. Notion of diversification reward is missing in the description of the proximal policy optimization.
> >
> > A5. Following your suggestion, we have added the graphical representation of our framework in Appendix A of the revised draft. The notion of diversification reward was also added to the description of proximal policy optimization in the last equation of Section 3.2.
> >
> > Q6. ReduMIS [2] is nowadays dominated by more efficient kernelization/heuristic techniques for solving MIS. It is suggested to replace ReduMIS with solvers of better performance, e.g., OnlineMIS [4].
> >
> > A6. Our implementation of ReduMIS [3] incorporates the recent graph kernelization technique by Hespe et al. [5]. We believe our choice of implementation is a strong baseline due to the following facts:
> >
> > 1) Hespe et al. [5] report better performance than Chang et al. [6] under incorporating their kernelization technique into ReduMIS.
> > 2) Chang et al. [6] in turn report better performance than OnlineMIS [4] under incorporating their kernelization technique into ARW [7].
> > 3) ARW [7] itself was shown to underperform compared to ReduMIS in the original ReduMIS paper [2].
> >
> > We have made changes in our revised draft to highlight this fact. In Section 4, first paragraph, 17th line, we state: “Note that we use the implementation of ReduMIS equipped with graph kernelization proposed by Hespe et al. 2019.”. Since specification on the state-of-the-art MIS solver can be debatable, we removed any claims on our implementation to be the state-of-the-art.
> >
> > Q7. It is suggested to compare with the supervised learning framework of Li et al. [8].
> >
> > A7. Following your suggestion, we include the comparison to Li et al. [8] in Appendix C of the revised draft. From the experiments, one observes that our framework outperforms the supervised learning framework when given a similar computational budget. We also refer our response A2 for R2.
> >
> > Q8. It is suggested to compare with the SMT solver Z3 [9] that was used for solving the MIS problem by Li et al. [8].
> >
> > A8. Thank you for the constructive feedback. We would like to point out that Z3 has been used as a SAT solver by Li et al. [8], instead of being used as an MIS solver. Specifically, they compare the performance of Z3 for solving the SAT problems with other MIS solvers for solving the MIS problems (reduced from the same SAT problems). We believe that using Z3 for solving arbitrary MIS problem is non-trivial, e.g., Li et al. [8] states in Section 5.2, 2nd paragraph, 11th line of their paper as follows: “Note that Z3 directly solves the SAT problem and cannot solve any transformed MIS problem on the SATLIB dataset”.
> >
> > [2] Sebastian Lamm, Peter Sanders, Christian Schulz, Darren Strash and Renato F. Werneck, “Finding Near-Optimal Independent Sets at Scale”, In Journal of Heuristics, 2017
> > [3] http://algo2.iti.kit.edu/kamis/
> > [4] Jakob Dahlum, Sebastian Lamm, Peter Sanders, Christian Schulz, Darren Strash and Renato F. Werneck, “Accelerating Local Search for the Maximum Independent Set Problem”, In International symposium on experimental algorithms. Springer, 2016
> > [5] Demian Hespe, Christian Schulz and Darren Strash, “Scalable Kernelization for Maximum Independent Sets”, In JEA, 2019
> > [6] Lijun Chang, Wei Li and Wenjie Zhang. “Computing a near-maximum independent set in linear time by reducing-peeling.” In SIGMOD, 2017.
> > [7] Diogo V. Andrade, Mauricio G.C. Resende and Renato F. Werneck “Fast Local Search for the Maximum Independent Set Problem”, In Journal of Heuristics, 2012
> > [8] Zhuwen Li, Qifeng Chen and Vladlen Koltun, “Combinatorial Optimization with Graph Convolutional Networks and Guided Tree Search”, In NeurIPS, 2018
> > [9] Leonardo Mendonça de Moura and Nikolaj Bjørner, “Z3: An efficient SMT solver”, In TACAS, 2008

---

### Official Review · AnonReviewer1 · 2019-10-23
**Official Blind Review #1**

**Rating:** 6

**Review:**

The paper introduces auto-deferring policies (ADPs) for deep reinforcement learning (RL). ADPs automatically stretching or shrinking their decision process, in particular, deciding whether to finalize the value of each vertex at the current stage or defer to determine it at later stages. ADPs are evaluated on maximum independent set problems.

The paper is in principle well written and structured. Some statements of the paper appear a little bit too strong. For instance, saying that deep RL approaches "can automatically learn the design of a good solver without using any sophisticated knowledge or hand-crafted heuristic specialized for the target problem" is misleading as thee designer of the RL setup is putting a lot of knowledge into the design. Likewise the statement "without any human guidance" is not true, at least at the current stage. It would be great to acknowledge this by softening this statement.

The basic idea is also fine, learning to expand or not nodes in the current fringe of the combinatorial solver. However, being an informed outsider, I would like to understand more why encoding NP-hard problem using an (discrete, finite) MDP, which is rather efficient to solve, is a good idea. Moreover, while the focus on MIS is justified in the paper, showing the (potential) benfit on other tasks such as TSP is useful to convince the reader that ADPs apply across different problem classes. Without, it is not clear whether ADPs work fine on other problem classes (even if one may expect that this is the case).

Anyhow, the main idea of implementing two independent agents/networks to implement a reward signal that rewards deviation is interesting. Moreover, the experimental results show that the approach works. It actually manages to be on par with ReduMIS. However, it does not really improve upon this well-known heuristic. Hence some experiments across different problems would really be beneficial. Without, it is just interesting to see that RL can also come up with good heuristics but the existing heuristic already works pretty well.

To sum up, a very nice idea that shows promise but experiments on other problems is missing for a complete picture. Also some statements should be soften.

**Experience Assessment:**

I have published one or two papers in this area.

**Review Assessment: Checking Correctness Of Derivations And Theory:**

I assessed the sensibility of the derivations and theory.

**Review Assessment: Checking Correctness Of Experiments:**

I carefully checked the experiments.

**Review Assessment: Thoroughness In Paper Reading:**

I read the paper thoroughly.

---

> ### Author Response · Authors · 2019-11-14
> **Response to R1**
>
> Thank you for your valuable efforts and time spent on reading our paper. We believe our framework proposes an interesting idea (as pointed out by you and R3) with promising empirical results (as pointed out by R2 and R3), based on clear writing (as pointed out by you and R2). In the revised draft, we added a description of related works (in Section 2), a graphical illustration of our framework (in Appendix A) and additional experiments (in Appendix B, C, D) to incorporate all reviewers’ comments. Any modifications from the original draft are highlighted by “red” texts. We think the revised draft has much improved by clarifying and improving the contributions of our work, thanks to your comments. Our responses to all your questions are provided below.
>
> Q1. Some statements of the paper appear a little bit too strong.
>
> A1. Thank you for the suggestion. In the revised draft, we softened several sentences, e.g.,
>
> - Abstract, 4th line:
> (before revision) Recently… they can automatically learn the design of a good solver without using any sophisticated knowledge or hand-crafted heuristic specialized for the target problem.
> (after revision) Recently ... they can automatically train a good solver while relying less on sophisticated domain knowledge of the target problem.
>
> - Introduction, 3rd paragraph, 4th line:
> (before revision) Then the corresponding agent can be trained (without any human guidance) based on existing training schemes of DRL.
> (after revision) Then, the corresponding agent can be trained based on existing training schemes of DRL, e.g., ...
>
> Q2. Why is encoding NP-hard problem using a (discrete, finite) Markov decision process (MDP), which is rather efficient to solve, is a good idea?
>
> A2. Thank you for raising a very important question. While a NP-hard problem is hard to solve exactly, it is often good enough to use a heuristic for obtaining an approximate solution under a reasonable amount of time. This motivates to express heuristic as a MDP with “finite” episode length, which is able to be optimized for generating high-quality solutions under a given time constraint.
>
> Furthermore, the MDP can be used to learn new problems while relying less on problem-specific domain knowledge, which is often expensive to acquire. To illustrate, we introduce the prize collecting maximum independent set (PCMIS) problem in Appendix B of the revised draft. In the PCMIS problem, the “hard” constraints of a MIS, i.e., forbidding two adjacent vertices, are replaced by a “soft” penalty term. For this problem, existing MIS heuristics are not applicable without substantial changes relying on problem-specific knowledge provided by experts. On the contrary, our MDP can be trained to solve the problem with a very small modification, i.e., excluding the clean-up phase.
>
> Q3. Experiments on problems other than the maximum independent set (MIS) problem are missing for a complete picture.
>
> A3. Following your suggestion, we applied our framework to other problems in Appendix B of the revised draft, i.e., the maximum weighted independent set (MWIS) problem and the PCMIS problem. Such choices are due to our decision to focus on the MIS-like problems in the revised draft, following the suggestion of R3. In the new experiments, our framework shows its general applicability by achieving better performance than the generic integer programming solver, i.e., CPLEX. Note that ES and ReduMIS are not applicable to the considered variants of the MIS problem, which highlights the advantage of our scheme.

---

### Official Review · AnonReviewer2 · 2019-10-23
**Official Blind Review #2**

**Rating:** 6

**Review:**

The paper proposes a Deep RL approach called Auto-Deferring Policy (ADP) to learning a policy for constructing solutions for the Maximum Independent Set (MIS) problem. Rather than constructing a solution one variable per episode step, the policy can make decisions about multiple variables per step, as well as defer decisions to later steps. At each step the MIS constraints of a valid solution are checked and decisions that violate the constraints are reverted, and any additional decisions that are automatically implied by the decisions so far and the MIS constraints are taken. The policy and value function are parameterized as a graph convolutional network to make use of the graph structure of the problem. A diversity regularizer that encourages the policy to generate diverse final solutions is used for training. Results show that the approach is able to match the objective value of the state-of-the-art solvers on several datasets, and is able to outperform S2V-DQN on several datasets when neither approach is trained on them.

Pros:
- Extensive experiments have been done on several datasets, both synthetic and real. The ablation and generalization experiments are particularly valuable for getting insight into the performance of the algorithm.
- Comparison to CPLEX and ReduMIS strengthens the results.
- The paper is reasonably well-written and easy to follow. Figures 2 and 3 are especially useful for quickly understanding key ideas.

Cons:
- The update and clean-up phases inject significant domain knowledge about MIS into the policy. At a high level the idea is similar to unit propagation in Boolean Satisfiability (SAT) solvers or domain propagation in Mixed Integer Programming (MIP) solvers, i.e., the hard constraints of the problem can be used to make decisions in the search for a solution. Both SAT and MIP solvers have been shown to rely on it to significantly improve their search. Improvements over S2V-DQN could be due to this extra built-in domain knowledge, rather than any improvements related to learning. For a fairer comparison, either this knowledge should be removed from ADP as an additional ablation study, or it should somehow be given to S2V-DQN as well. Without such a comparison, it is not clear that the improvements are really due to the new ideas like auto-deferring, diversity regularizer, etc.
- A comparison to Li, Chen, and Koltun, NeurIPS’18 is needed since they have shown strong results for MIS and they have made the code available online. Although their approach is supervised learning, it would still be useful to assess an RL approach’s performance relative to SL.


Additional comments:
- It is not clear what encourages the policy to defer decisions, rather than making all the {0,1} decisions for all variables in one step or a small number of steps. Does this behaviour of the policy arise naturally via training, or does some regularization need to be applied to ensure that the policy doesn’t prematurely fix all variables? For example, as a way to avoid/reduce variance in the reward signal, the policy may learn to fix all variables in just one step -- does this issue arise?

- The abstract says "The reported performance of our generic DRL scheme is also comparable with that of the state-of-the-art solvers specialized for MIS, e.g., ADP outperforms them for some graphs with millions of vertices." Can you please point out the specific results in the paper that this sentence is referring to?

- Comparison to CPLEX:  The paper mentions that “we observe that our algorithm outperforms the CPLEX solver on ER-(100, 200) and ER-(400, 500) datasets, while consuming a smaller amount of time”, and “It is remarkable to observe that ADP outperforms the CPLEX solver on both datasets under reasonably limited time.” Note that CPLEX not only optimizes the objective, but also proves a bound on the objective, while ADP only does the former. So this is not a fair comparison. A fairer comparison would be to set CPLEX hyperparameters to give higher priority to optimizing the objective and then measure the time CPLEX took to first reach a particular objective value rather than the time to “solve” an instance. It can be the case that a given objective value is achieved quickly but then proving a bound requires much longer. Also there is no mention of the optimality gap used as a stopping criterion for CPLEX.

- One potential advantage of making one decision per step is that the credit assignment problem may be simpler compared to making many decisions simultaneously. It would be insightful to explore this tradeoff in more detail.

- The acronym ADP is somewhat well-known in the optimization community as Approximate Dynamic Programming (http://adp.princeton.edu/), so it would be helpful to use a different one.

**Experience Assessment:**

I have published one or two papers in this area.

**Review Assessment: Checking Correctness Of Derivations And Theory:**

I assessed the sensibility of the derivations and theory.

**Review Assessment: Checking Correctness Of Experiments:**

I carefully checked the experiments.

**Review Assessment: Thoroughness In Paper Reading:**

I read the paper at least twice and used my best judgement in assessing the paper.

---

> ### Author Response · Authors · 2019-11-14
> **Response to R2 (1/2)**
>
> Thank you for your valuable efforts and time spent on reading our paper. We believe our framework proposes an interesting idea (as pointed out by R1 and R3) with promising empirical results (as pointed out by you and R3), based on clear writing (as pointed out by you and R1). In the revised draft, we added a description of related works (in Section 2), a graphical illustration of our framework (in Appendix A) and additional experiments (in Appendix B, C, D) to incorporate all reviewers’ comments. Any modifications from the original draft are highlighted by “red” texts. We think the revised draft has much improved by validating the empirical fairness and clarifying the intuition behind our framework, thanks to your comments. Our responses to all your questions are provided below.
>
> Q1. The proposed framework uses hard constraints of the maximum independent set (MIS) problem, which makes it unfair to compare with S2V-DQN [1].
>
> A1. We run the code [2] provided by the authors of S2V-DQN, which also utilizes similar hard constraints. Hence, we think our empirical comparison is fair. To explain in more detail, we first point out that S2V-DQN solves the minimum vertex cover problem (identical to the MIS problem). During its greedy selection of vertices, any vertices that are already fully covered by the intermediate vertex cover are automatically excluded. Such a process is very much similar to our use of hard constraints.
>
> Q2. Comparison with the supervised learning framework proposed by Li et al. [3] is needed since they also solve the MIS problem.
>
> A2. Following your suggestion, we include the comparison to Li et al. [3] in Appendix C of the revised draft. From the experiments, one can observe that our reinforcement learning framework outperforms the supervised learning framework when given a similar computational budget. Here, we remark that Li et al. [3] additionally introduced the idea of combining local search and graph reduction between intermediate steps. Although such an idea is applicable to both the supervised and reinforcement learning frameworks, we disable them in our experiments for a fair comparison, i.e., to use a similar amount of domain knowledge. We found that the strong results reported by Li et al. [3] can be reproduced when the local search and graph reduction are added to the supervised learning framework.
>
> [1] Elias B. Khalil, Hanjun Dai, Yuyu Zhang, Bistra Dilkina and Le Song, “Learning combinatorial optimization algorithms over graphs”, In NeurIPS, 2017
> [2] https://github.com/Hanjun-Dai/graph_comb_opt
> [3] Zhuwen Li, Qifeng Chen and Vladlen Koltun, “Combinatorial Optimization with Graph Convolutional Networks and Guided Tree Search”, In NeurIPS, 2018

---

> > ### Author Response · Authors · 2019-11-14
> > **Response to R2 (2/2)**
> >
> > Q3. Is there a case where the policy learns to fix all variables in just one step? It is not clear what encourages the policy to defer decisions.
> >
> > A3. Thank you for your interesting question. Empirically, the policy was not observed to fix all variables in just one step. Instead, it usually takes more than 30 steps to fix all variables in all our experiments. Our policy learns to defer decisions through higher rewards coming from better solving the problem. Intuitively, the deferring of decisions on vertices helps to solve the MIS problem by prioritizing making decisions on the “easier” vertices first. Whenever easier decisions are made, the surrounding values of “hard” vertices are fixed. Then, the decisions on “hard” vertices become easier to be made. For example, when a “hard” vertex becomes surrounded by “easy” vertices that were excluded from the final independent set in the previous iteration, it should be included in the independent set. In Section 3, the first paragraph, the 12th line of the revised draft, we added an explanation of such intuition for a better understanding of readers.
> >
> > Q4. Please point out specific results on outperforming the existing solver.
> >
> > A4. Thank you for your suggestion. In the revised draft, we have modified the 15th line (last sentence) of the abstract to point out the specific results. It is now stated as follows: “The reported performance of our generic DRL scheme is also comparable with that of the existing solvers for MIS, e.g., AutoDP outperforms them for the Barabasi-Albert graph with two million vertices.”. To avoid any confusion, we would also like to remark that the same statements are mentioned in the last paragraph of Section 1 and the last paragraph of Section 4.3 of both the original and the revised draft.
> >
> > Q5. Hyper-parameters of CPLEX should be adjusted for a fairer comparison.
> >
> > A5. To address your concerns, we tried setting the hyper-parameter of CPLEX to prioritize the optimization of objective in various ways. For your information, we report the performance of CPLEX under varying the optimality gap used for the stopping criterion, as in the last paragraph of Appendix E of the revised draft. However, we found that such a change of hyper-parameter tends to degrade in performance with marginal speedups. Therefore, we still keep the current settings of CPLEX in the revised draft.
> >
> > Nevertheless, to incorporate your comments, we have revised the paper to clarify this point for the readers. Specifically, we added the following description as a footnote in the first paragraph of Section 4: “Note that CPLEX is also able to prove an upper bound of the solution for the MIS problem.” We also describe our algorithm to “achieve a better objective” rather than to “outperform” over CPLEX.
> >
> >
> > Q6. It would be insightful to explore the tradeoff between making many decisions and better credit assignment.
> >
> > A6. Thank you for the interesting question. We did not observe any credit assignment problem even when training on quite large graphs, e.g., a graph with 500 vertices requires action with 500 dimensions. Instead, S2V-DQN (which makes one-dimensional decision per step) suffers more from unstable training in large graphs. We think this is due to the long episode length required for the execution of S2V-DQN (known as the temporal credit assignment problem [4]). We still think your question is very interesting and hope to explore this point in the future.
> >
> > Q7. Acronym “ADP” is already known in the optimization community as Approximate Dynamic Programming.
> >
> > A7. Thank you very much for your information. Following your suggestion, we have changed our acronym to “AutoDP”.
> >
> > [4] Richard S. Sutton, “Learning to predict by the methods of temporal differences”, In Springers, 1988

---

### Author Response · Authors · 2019-11-15
**Common response to all reviewers: summary of revision**

Dear all reviewers,

First of all, we appreciate your valuable time and effort to review our paper.

In the revised draft, we enhanced our manuscript with a substantial amount of materials, including:

- a separate description of related works (Section 2), suggested by R3
- a graphical illustration of our framework (Appendix A), suggested by R3
- experimental results for two more combinatorial problems (Appendix B), suggested by R2 and R3
- comparison with the state-of-the-art supervised learning (Appendix C), suggested by R1 and R3
- evaluation of CPLEX with different hyperparameters (Appendix D), suggested by R1

Any modifications from the original draft are highlighted by “red” texts.

Thank you very much.

---

### Decision · Program_Chairs · 2019-12-19

**Decision:**

Reject

**Comment:**

This paper proposes a new way to formulate the design of the deep reinforcement learning that automatically shrinks or expands decision processes.

The paper is borderline and all reviewers appreciate the paper and gives thorough reviews. However, it not completely convince that it is ready publication.

Rejection is recommended. This can become a nice paper for next conference by taking feedback into account.